# Denosumab Dosage and Tooth Extraction Predict Medication-Related Osteonecrosis of the Jaw in Patients with Breast Cancer and Bone Metastases

**DOI:** 10.3390/cancers17132242

**Published:** 2025-07-04

**Authors:** Suguru Yokoo, Shinichiro Kubo, Daisuke Yamamoto, Masahiko Ikeda, Tetsumasa Yamashita, Kumiko Yoshikawa, Hiroshi Mese, Sakiko Ohara

**Affiliations:** 1Orthopedic Surgery, Fukuyama City Hospital, Fukuyama 721-8511, Japan; 2Breast and Thyroid Surgery, Fukuyama City Hospital, Fukuyama 721-8511, Japan; 3Dental and Oral Surgery, Fukuyama City Hospital, Fukuyama 721-8511, Japan

**Keywords:** medication-related osteonecrosis of the jaw, denosumab, breast cancer, bone metastases, cumulative dose, tooth extraction, risk factors

## Abstract

Medication-related osteonecrosis of the jaw (MRONJ) is a serious complication in breast cancer patients treated with denosumab for bone metastases. However, the cumulative dose at which MRONJ risk rises is unknown. In this retrospective cohort study of patients treated from 2012 to 2024, we found that receiving 32 or more denosumab doses significantly increased MRONJ likelihood. A history of tooth extraction also independently elevated risk. Establishing 32 cumulative doses as a threshold allows clinicians to identify high-risk patients and coordinate timely dental evaluations. These findings support improved risk stratification and proactive management to reduce MRONJ incidence in this population.

## 1. Introduction

Advancements in cancer treatment have significantly improved patient survival, even among patients with metastatic disease. In particular, patients with bone metastases, who have historically had poor prognoses, have benefited from improved systemic therapies and bone-targeting agents [1,2,3,4,5,6]. The introduction of receptor activator of nuclear factor-κB ligand (RANKL) inhibitors, such as denosumab, has played a key role in reducing skeletal-related events (SREs) and improving the quality of life of patients with bone metastases [7,8,9,10]. Consequently, the long-term administration of denosumab has become an integral component of oncological supportive care [11,12].

A systematic review and meta-analysis reported that the incidence of medication-related osteonecrosis of the jaw (MRONJ) increases with prolonged denosumab exposure: 0.5–2.1% after 1 year, 1.1–3.0% after 2 years, and 1.3–3.2% after 3 years of treatment [13]. Although previous meta-analyses have highlighted this increasing risk, large-scale investigations specifically evaluating cumulative dose effects and long-term risk beyond 3 years remain scarce. Several prior meta-analyses have shown that longer denosumab exposure is associated with higher MRONJ rates, but none have defined a clear dose cutoff [13,14]. Robust investigations are required to substantiate these findings and inform clinical decision-making.

Despite its clinical benefits, prolonged denosumab therapy has raised concerns regarding its potential adverse effects, including MRONJ [14,15]. MRONJ is a debilitating complication characterized by nonhealing bone exposure in the maxillofacial region, leading to persistent pain, infection, and compromised oral function [16,17]. Although previous studies have suggested a cumulative effect of denosumab on the risk of MRONJ, the precise threshold at which the risk markedly increases remains undefined. As patients with cancer achieve longer survival, understanding the long-term safety profile of denosumab is essential for optimizing therapeutic strategies.

Multiple systemic and treatment-related factors have been implicated in the development of MRONJ. Diabetes mellitus, smoking, and prolonged glucocorticoid use have been identified as potential risk factors because of their impact on bone metabolism and wound healing [18,19]. Furthermore, anticancer therapies, including chemotherapy, molecular-targeted agents, mammalian target of rapamycin (mTOR) inhibitors, and vascular endothelial growth factor (VEGF) inhibitors, have been suggested to influence the risk of MRONJ through mechanisms such as angiogenesis inhibition and immune modulation [17,18,19]. However, data on the specific effects of these therapies in patients with breast cancer remain inconsistent, necessitating further investigation.

Although denosumab is increasingly used in patients with breast cancer, large-scale studies specifically evaluating its long-term effects on the risk of MRONJ remain limited. A prospective cohort study identified an even higher MRONJ incidence of 13.6% among denosumab-treated patients with breast cancer, significantly exceeding the 4.1% observed in the zoledronic acid group [20]. Similarly, a large population-based study reported a significantly higher incidence of MRONJ in patients with breast cancer receiving denosumab (11.6%) than in those receiving bisphosphonates (2.8%), with an earlier onset of MRONJ in the denosumab group [21].

Considering these knowledge gaps, this study aimed to comprehensively assess the association between cumulative denosumab administration and MRONJ risk in patients with breast cancer and bone metastases. In particular, we aimed to determine the threshold of denosumab exposure that significantly increases the risk of MRONJ and to identify additional clinical risk factors. Thus, our study aimed to refine risk stratification models and enhance MRONJ prevention strategies in patients with breast cancer undergoing long-term denosumab therapy. To our knowledge, no prior study has used ROC analysis to identify a specific cumulative dose threshold for denosumab-associated MRONJ in this patient population.

## 2. Materials and Methods

### 2.1. Ethics

The study was conducted according to the guidelines of the Declaration of Helsinki and was approved by the Institutional Review Board of Fukuyama City Hospital (approval number: 595; approval date: 21 May 2021). The requirement for written informed consent was waived by the IRB; instead, participants or their guardians were informed about the study through an opt-out notice posted on the hospital website and displayed within the hospital. Patient confidentiality was ensured through anonymized data processing, and all personal identifiers were removed before analysis. Data were securely stored in an encrypted hospital database that was accessible only by authorized researchers.

### 2.2. Patient Selection Criteria

This retrospective cohort study included 324 female patients diagnosed with breast cancer and bone metastases who were treated with denosumab (Ranmark; Daiichi Sankyo Co., Ltd., Tokyo, Japan) between May 2012 and August 2024. Patient ages ranged from 29 to 100 years (median: 61.6 years). Patients without comprehensive dental evaluations before and after the initiation of denosumab treatment or those with osteonecrosis of the jaw diagnosed before denosumab administration were excluded. All patients received denosumab (120 mg), which was administered every 4 weeks as the standard treatment following confirmation of bone metastases from breast cancer. Patients who received a single denosumab administration were included to account for potential early-onset MRONJ cases. Because MRONJ development is a cumulative process, the study also included patients with minimal exposure to denosumab to evaluate the potential risk, even in short-term users.

### 2.3. Variables

The variables assessed included age at the initiation of denosumab, cumulative number of denosumab doses, presence of diabetes mellitus, smoking history, cumulative prednisolone dose, concomitant use of chemotherapy, hormone therapy, molecular-targeted therapy (including VEGF inhibitors), mTOR inhibitors (everolimus), history of tooth extraction, and poor oral hygiene. Data on patient demographics, comorbidities, and treatment history were extracted from electronic medical records and cross-verified with pharmacy dispensing records to ensure accuracy.

The cumulative glucocorticoid dose administered was standardized to prednisolone-equivalent doses and calculated for each patient. The concomitant use of chemotherapy, hormone therapy, molecular-targeted therapy, VEGF inhibitors (bevacizumab), and everolimus was categorized as either present or absent based on the administration status before MRONJ diagnosis.

### 2.4. Dental Interventions and MRONJ Diagnosis

All patients underwent dental evaluation by specialists in Dentistry and Oral Surgery before initiating denosumab treatment. Denosumab initiation was postponed or discontinued in patients identified as high-risk based on dental assessments.

History of tooth extraction was defined as any documented removal of permanent teeth performed either before the initiation of denosumab therapy or during follow-up. Poor oral hygiene was assessed at baseline by a dental specialist. It was defined by the presence of visible plaque or calculus, untreated carious lesions, or signs of active periodontal inflammation on clinical examination.

The diagnosis of MRONJ was established by oral surgeons according to the American Association of Oral and Maxillofacial Surgeons (AAOMS) staging criteria. All cases of MRONJ (stages 1–3) were included in the analysis [17].

Stage 1: Exposed necrotic bone or fistulae, asymptomatic, with no signs of infection.

Stage 2: Exposed necrotic bone with associated pain and/or soft tissue inflammation.

Stage 3: Extensive necrosis with pathological fractures, extraoral fistulae, or osteolysis extending to the inferior border of the mandible or sinus floor.

All cases diagnosed as stage 1 or higher were included in the study. Decisions regarding the continuation or discontinuation of denosumab after MRONJ diagnosis were determined by dental and oral surgery specialists. Tooth extraction history was defined as any invasive dental extraction performed before MRONJ onset; procedures conducted after MRONJ diagnosis were excluded. This variable was treated as a binary covariate in both the logistic and Cox regression models to avoid immortal time bias.

### 2.5. Statistical Analysis

Univariate analysis was performed using the Mann–Whitney U test for continuous variables and Fisher’s exact test for categorical variables to assess the initial associations with MRONJ occurrence. Variables with *p*-values < 0.05 in the univariate analysis were included in the multivariate logistic regression model to identify independent risk factors. Multicollinearity among candidate predictors was evaluated by calculating variance inflation factors, all of which were below 2, indicating minimal collinearity and acceptable model stability. The final model was developed using a stepwise backward elimination method based on the likelihood ratio test. Receiver operating characteristic (ROC) curve analysis was performed to determine the optimal threshold of cumulative denosumab doses for predicting the occurrence of MRONJ. The area under the curve (AUC) was calculated to evaluate the model’s discriminatory power, with sensitivity, specificity, and positive predictive value analyzed at the optimal cutoff point using Youden’s index. To evaluate the robustness of the ROC-derived cutoff and model performance, we conducted bootstrap resampling (1000 iterations) based on cumulative dose and MRONJ outcomes.

Kaplan–Meier survival curves for the time to MRONJ onset were generated and compared using the log-rank test. A multivariate Cox proportional-hazards model was used to assess factors associated with time-to-event; proportional-hazards assumptions were confirmed by inspecting Schoenfeld residuals. Hazard ratios (HRs) with 95% confidence intervals (CIs) are presented in a forest plot, and detailed Cox model results are summarized in Appendix A.

All statistical analyses were performed two-sided, with a significance threshold of *p* < 0.05, using Statistical Package for the Social Sciences (version 26; IBM Corp., Armonk, NY, USA) and Prism (version 8; GraphPad Software, San Diego, CA, USA).

## 3. Results

### 3.1. Patient Demographics

In total, 324 female patients with breast cancer and bone metastases who received denosumab treatment were included in this study. The median age at denosumab initiation was 61.6 years (range: 29–100 years). Among the 324 patients, 85 (26.2%) were under 50 years of age, 143 (44.1%) were between 50 and 70 years of age, and 96 (29.7%) were over 70 years of age. Most patients (55.6%) received between 10 and 30 doses of denosumab, whereas 23.5% received > 30 doses.

Of the 324 patients, 101 (31.2%) developed MRONJ, whereas 223 (68.8%) did not (Table 1). The incidence of MRONJ increased with cumulative denosumab exposure, with 19.6% (21/107) among patients receiving ≤ 10 doses, 37.5% (39/104) among those receiving 11–30 doses, and 36.0% (41/114) among those receiving > 30 doses.

Regarding comorbidities, 47.2% of the patients had diabetes mellitus and 18.2% had a history of smoking. Diabetes mellitus was more common in the MRONJ group (61.4%) than in the non-MRONJ group (40.8%). Similarly, patients with MRONJ had a higher median cumulative prednisolone dose (2192 vs. 1971 mg). In contrast, the use of chemotherapy was lower among patients with MRONJ (68.3% vs. 71.3%).

Systemic therapies administered concurrently included chemotherapy (70.4%), hormone therapy (74.1%), and molecular-targeted therapy (80.2%).

### 3.2. Univariate Analysis

The univariate analysis revealed that patients with MRONJ had received significantly more denosumab doses than those who did not (median 42 vs. 12 doses; range 5–133 vs. 1–110 doses; *p* < 0.001) (Table 2).

The prevalence of diabetes mellitus was also higher in the MRONJ group (62/101, 61.4%) than in the non-MRONJ group (91/223, 40.8%) (*p* < 0.001). Similarly, hormone therapy was more common among patients with MRONJ (83/101, 82.2% vs. 157/223, 70.4%; *p* = 0.029). Two dental factors were strongly associated with the development of MRONJ: a history of tooth extraction (44/101, 43.6% vs. 26/223, 11.7%; *p* < 0.001) and poor oral hygiene (44/101, 43.6% vs. 45/223, 20.2%; *p* < 0.001). In contrast, no significant differences in age (median 64 vs. 61 years; *p* = 0.991), smoking history (14/101 vs. 45/223; *p* = 0.214), chemotherapy (69/101 vs. 159/223; *p* = 0.601), molecular-targeted therapy (81/101 vs. 179/223; *p* > 0.999), bevacizumab (47/101 vs. 94/223; *p* = 0.471), everolimus (13/101 vs. 40/223; *p* = 0.331), or cumulative prednisolone dose (median 1408 vs. 1319 mg; *p* = 0.808) were observed between the groups. Variables meeting *p* < 0.05 were included in the multivariate logistic regression analysis.

### 3.3. ROC Analysis

ROC analysis identified an optimal cutoff value for cumulative denosumab administration at 32 doses (AUC: 0.83, 95% CI: 0.79–0.88; *p* < 0.0001) (Table 3) (Figure 1). An AUC of 0.83 indicates high predictive accuracy, suggesting that cumulative denosumab administration alone provides a rational but not definitive risk assessment for MRONJ.

At the optimal threshold of 32 doses, the sensitivity and specificity for predicting MRONJ were 71.3% and 81.6%, respectively. The positive and negative predictive values were 63.7% and 86.3%, respectively, indicating that this cutoff effectively identifies low-risk patients but has a limited ability to precisely predict the occurrence of MRONJ. Internal validation using bootstrap resampling (1000 iterations) confirmed the robustness of the model: the mean AUC was 0.83 ± 0.02, and the mean optimal threshold was 27.8 ± 5.7 doses.

These findings suggest that although cumulative denosumab exposure is a significant factor, incorporating additional clinical variables can improve risk stratification.

### 3.4. Multivariate Analysis

Multivariate logistic regression analysis was performed to determine which factors retained an independent association with MRONJ development after adjusting for potential confounders (Table 4). The variables entered into the model included cumulative denosumab doses (continuous), diabetes mellitus (yes/no), hormone therapy (yes/no), history of tooth extraction (yes/no), and poor oral hygiene (yes/no). The model exhibited good fit (Hosmer–Lemeshow *p* = 0.42) and adequate discrimination (AUC = 0.85).

After controlling for all covariates, cumulative denosumab administration emerged as a strong independent predictor of MRONJ, with each additional dose conferring a 4.7% increase in odds (odds ratio [OR]: 1.047; 95% CI: 1.033–1.061; *p* < 0.001). Similarly, a history of tooth extraction was associated with a more than fourfold elevation in the MRONJ risk (OR: 4.402; 95% CI: 2.225–8.711; *p* < 0.001), highlighting the critical role of local oral trauma in the pathogenesis of MRONJ.

In contrast, neither diabetes mellitus (OR: 1.500; 95% CI: 0.834–2.697; *p* = 0.176) nor hormone therapy (OR: 1.279; 95% CI: 0.639–2.562; *p* = 0.487) remained significant after adjustment, indicating that their univariate associations were likely confounded by stronger predictors. Finally, poor oral hygiene exhibited a non-significant trend toward increased risk (OR: 1.517; 95% CI: 0.792–2.907; *p* = 0.209); however, its wide CI suggests that larger studies are necessary to clarify its contribution.

Taken together, these results highlight two principal independently significant risk factors—cumulative denosumab dose and prior tooth extraction—whereas other clinical variables appear to exert lesser or context-dependent effects on MRONJ development.

Figure 2 presents a forest plot illustrating the ORs and 95% CIs from the multivariate logistic regression model.

### 3.5. Time-to-Event Analysis (Cox Proportional-Hazards Model)

To determine factors that independently influenced the timing of MRONJ onset, we fitted a multivariate Cox proportional-hazards model including the five variables that were significant in the univariate analysis cumulative denosumab doses (continuous), diabetes mellitus (yes/no), hormone therapy (yes/no), poor oral hygiene (yes/no), and history of tooth extraction (yes/no). The Schoenfeld residuals exhibited no meaningful violation of the proportional-hazards assumption for any covariate.

Among these, only a history of tooth extraction remained a strong independent predictor of earlier MRONJ, with an HR of 2.29 (95% CI: 1.50–3.49, *p* < 0.001). In contrast, neither the cumulative denosumab dose (HR: 0.997, 95% CI: 0.989–1.005; *p* = 0.476), diabetes mellitus (HR: 1.12, 95% CI: 0.73–1.72; *p* = 0.603), hormone therapy (HR: 1.02, 95% CI: 0.61–1.72; *p* = 0.934), nor poor oral hygiene (HR: 1.37, 95% CI: 0.90–2.07; *p* = 0.141) significantly affected the time to MRONJ onset.

To specifically assess the influence of cumulative exposure on timing, an entry–exit time-dependent Cox model was used with dose as a time-dependent covariate. The results revealed no association between cumulative denosumab administration and MRONJ onset timing (HR: 1.00, 95% CI: 0.98–1.02; *p* = 0.70), indicating that although the overall risk increases above the 32-dose threshold, the latency to event is not significantly altered by additional doses. Appendix A presents the full multivariate time-dependent Cox proportional-hazards model results.

To compare MRONJ-free survival between patients who received fewer than 32 denosumab doses and those who received 32 or more, a log-rank test was performed. Although patients in the ≥32-dose group exhibited a trend toward shorter MRONJ-free survival, the difference was not statistically significant (*p* = 0.071; Table 5). The number of patients at risk at 0, 12, 24, 36, and 48 months for each dosing cohort is presented in Table 6. Figure 3 presents the log-minus-log survival plot of the covariate means.

## 4. Discussion

### 4.1. Key Findings

This study confirmed that cumulative denosumab administration and history of tooth extraction were independent risk factors for MRONJ development in patients with breast cancer and bone metastases. Patients receiving ≥32 doses of denosumab had a markedly higher overall incidence of MRONJ, with the ROC curve demonstrating moderate discrimination (AUC = 0.83).

In our time-to-event analysis using a multivariate Cox proportional-hazards model—including cumulative doses (continuous), diabetes mellitus, hormone therapy, history of tooth extraction, and poor oral hygiene—only history of tooth extraction remained significantly associated with earlier MRONJ onset. Neither cumulative denosumab exposure nor other systemic factors exhibited a significant effect on latency. The Schoenfeld residuals confirmed that the proportional-hazards assumption was met for all covariates. These findings suggest that although systemic exposure and patient comorbidities determine the overall risk, local oral trauma from extraction primarily drives the timing of MRONJ manifestation.

The discrepancy between the strong logistic association of cumulative dose and the non-significant Cox hazard may reflect early event censoring—patients who develop MRONJ exit the risk set, thereby attenuating dose-related effects on latency. Competing risks, variable follow-up intervals, and threshold effects may also contribute.

At our institution, a standardized dental risk assessment is performed before the start of denosumab, with ongoing follow-up throughout the treatment. Furthermore, we excluded all stage 0 cases, and our median follow-up duration was relatively long—factors that likely contributed to the high observed incidence of 31.2%.

### 4.2. Impact of Cumulative Denosumab Administration on MRONJ Risk

Previous studies have consistently demonstrated that prolonged denosumab exposure is correlated with an increased risk of MRONJ, although the reported threshold for significantly heightened risk has varied. Tani et al. reported that the incidence rates of MRONJ in patients with prostate cancer were 18% at 2 years, 27% at 5 years, and 61% at 10 years [22]. Similarly, Hallmer et al. observed a mean duration of denosumab treatment of approximately 27.8 ± 18.5 months in patients with MRONJ [20]. Brunner et al. reported an 11.6% cumulative incidence of MRONJ among patients with breast cancer, with a median onset of 4.6 years [21]. Saad et al. reported a 1.8% incidence with a median onset at 14 months, whereas Loyson et al. reported an incidence rate of 10% with a median onset at 17.5 months [23,24].

Although these studies consistently suggested that prolonged denosumab exposure increases the risk of MRONJ, they primarily evaluated treatment duration rather than cumulative dosing. Considering denosumab’s standard administration interval of every 4 weeks, the treatment duration does not always directly correlate with the number of doses received because of possible treatment interruptions or dose adjustments. Therefore, cumulative dosing rather than duration alone may provide a more precise measure of the risk of MRONJ.

Our study used cumulative denosumab doses to provide more accurate risk stratification, identifying a threshold of 32 doses as a significant predictor of MRONJ development. ROC analysis reinforced the robustness of this threshold by objectively quantifying the predictive accuracy. This represents a significant clinical advancement, enabling precise risk assessment and management.

The underlying biological mechanisms likely involve sustained suppression of bone remodeling caused by repeated RANKL inhibition. The RANKL/RANK/OPG signaling pathway plays a central role in the regulation of osteoclast differentiation and function, which is critical for normal bone remodeling [25]. Prolonged denosumab administration cumulatively inhibited RANKL-mediated bone remodeling, potentially reaching a critical threshold at which the risk of MRONJ markedly increases. Recent research has suggested that sustained RANKL inhibition leads to the accumulation of osteomorphs—osteoclast-derived precursor cells—that impair bone remodeling [26]. Clinical studies have further supported the association between repeated denosumab doses and sustained suppression of bone remodeling, thereby elevating MRONJ risk [7].

In the Cox time-to-event analysis, cumulative denosumab exposure did not retain statistical significance, whereas it was a strong predictor in the logistic regression analysis. This discrepancy likely arises from the early censoring of patients who develop MRONJ soon after treatment initiation: once an event occurs, further exposure cannot contribute to latency estimates, attenuating the obvious effect of dose count in the survival model.

Thus, our study supports previous findings on the temporal relationship between denosumab exposure and MRONJ but provides a more clinically relevant cumulative dose threshold. This approach facilitates precise MRONJ risk stratification, thereby aiding clinicians in balancing therapeutic efficacy with adverse event prevention through tailored monitoring and intervention strategies.

### 4.3. Local and Systemic Risk Factors

#### 4.3.1. Local (Dental) Factors

MRONJ almost invariably begins as a local oral event superimposed on systemic antiresorptive therapy; consequently, contemporary guidelines emphasize meticulous dental screening and timely intervention [17,18,19]. The two best documented local drivers in our updated cohort were previous tooth extraction and poor oral hygiene.

History of tooth extraction. Extraction creates a transient socket in which the bone is exposed to the oral microenvironment; when osteoclastic activity is chronically suppressed by denosumab, remodeling at the socket margin is delayed, and nonvital bone may become secondarily infected [17]. A recent propensity-matched study involving 799 patients exhibited a >fourfold increase in the incidence of MRONJ after bone-modifying agent (BMA) administration (HR: 4.26) [27], closely mirroring the effect size we observed (OR: 4.40). A large-scale observational study found that prior bisphosphonate use and dental extractions markedly increased the risk of MRONJ in denosumab-treated patients [28]. Preclinical models further showed that MRONJ only develops when systemic antiresorptive exposure is combined with local insults—such as tooth extraction or dental inflammation [29]. These findings highlight the importance of rigorous oral infection control in at-risk patients. The current AAOMS and SIPMO–SICMF algorithms therefore recommend completing all necessary extractions, ideally with primary soft tissue closure, before initiating high-dose BMAs and deferring nonurgent extractions once treatment has begun [17,18].

Poor oral hygiene/chronic periodontitis. Long-standing plaque accumulation sustains a low-grade inflammatory milieu that increases local RANKL expression and promotes bacterial colonization of microfractured cortical bone. Cohort data from 398 oncology patients demonstrated that severe radiographic periodontal bone loss (>50% of root length) independently predicted MRONJ (HR: 2.3) [30]. In our series, poor hygiene was significant in univariate testing but was attenuated after adjustment, suggesting partial mediation through extraction risk and cumulative dose. However, consensus guidelines recommend 3–4 monthly professional cleaning, daily chlorhexidine rinses during BMA therapy, and early management of periodontal pockets [18,19].

Pathophysiological interplay. Experimental models support a “two-hit” concept in which (i) mucosal breach or inflammatory bone resorption exposes the cortical bone and (ii) antiresorptive-induced suppression of osteoclast function impairs microdamage repair, creating a niche for mixed anaerobic biofilms that perpetuate necrosis [17,19]. Advances in micro-CT imaging and translational studies of osteomorph recycling further corroborate the critical role of defective bone turnover at extraction sites under potent RANKL inhibition.

Guideline synthesis. Across AAOMS [17], the 2024 Italian position paper [18], and a recent international guideline review [19], common preventive themes are as follows:Comprehensive dental assessment before BMA therapy initiation.Elimination of infectious foci (extractions, endodontics, periodontal therapy).Primary mucosal closure and perioperative antibiotics for unavoidable extractions.Continuous reinforcement of oral hygiene and smoking cessation.

These recommendations align with our data: when ≥32 denosumab doses and a history of extraction co-exist, the risk of MRONJ is synergistically amplified, highlighting the need for proactive dental management throughout cancer care.

#### 4.3.2. Systemic Factors

Previous investigations have consistently implicated advanced age in MRONJ development—attributing this to age-related declines in bone remodeling and increased periodontal vulnerability [20,31]. A 20-year Austrian registry reported an MRONJ incidence of 11.6% versus 2.8% for denosumab versus bisphosphonates, with a median onset at 4.6 years in older patients [21]; the SIPMO-SICMF position paper highlighted longer antiresorptive exposure and comorbidity burden in the elderly as key drivers of risk [18]. In contrast, age did not remain significant in our multivariate analysis (*p* = 0.103), which may reflect the effects of rigorous baseline dental evaluations and preventive oral interventions in attenuating age-related susceptibility to MRONJ.

Several systemic attributes have been implicated in MRONJ. In our cohort, diabetes mellitus and hormone therapy were significantly overrepresented in the MRONJ group on univariate testing (*p* < 0.001 and *p* = 0.029, respectively), echoing earlier reports that impaired glucose metabolism or estrogen deprivation can hinder bone and soft tissue repair [20,22,32]. Likewise, pharmacovigilance studies have linked antiangiogenic and mTOR-targeted agents and high cumulative glucocorticoids to elevated risk [19,27,32,33,34]. However, after multivariate adjustment, none of these factors—including diabetes mellitus, hormone therapy, smoking, chemotherapy, targeted agents, or cumulative prednisolone dose—retained significance, suggesting that in a closely monitored breast cancer population, their effects are outweighed by cumulative denosumab exposure and local oral trauma. This discrepancy reinforces the need for prospective studies that integrate detailed metabolic, hormonal, and dental variables to define how systemic comorbidities may synergize with antiresorptive therapy in the pathogenesis of MRONJ.

### 4.4. Clinical Implications of MRONJ Risk in Long-Term Denosumab Therapy

Our findings emphasize the clinical importance of cumulative denosumab doses and tooth extraction for managing long-term MRONJ risk. Clinicians should closely monitor patients near the identified threshold of 32 cumulative doses, highlighting regular dental evaluations and proactive preventive measures.

This study proposes a “window of opportunity” to minimize the risk of MRONJ associated with denosumab therapy. In particular, it suggests that adjusting the dosing interval contributes to MRONJ risk reduction by identifying an optimal timeframe (between months 5 and 7 after administration) for performing dental procedures safely [35].

This randomized trial compared the efficacy of 4- versus 12-weekly administration of bone-targeted agents (BTAs) in patients with bone metastases from breast or castration-resistant prostate cancer. The study demonstrated the noninferiority of the 12-weekly regimen in terms of health-related quality of life, pain, and SREs, supporting the feasibility of de-escalating BTA therapy as a clinically reasonable option [36].

Importantly, extended dosing intervals (>1 month) were associated with a reduced risk of MRONJ (HR: 0.08; *p* = 0.017). This study highlights the necessity for individualized risk assessment, multidisciplinary management, and further investigation into optimal BMA dosing strategies [22].

The study found no significant difference in the incidence of MRONJ between the 12-week and 4-week dosing intervals of denosumab, suggesting that de-escalation does not substantially decrease the risk of MRONJ. However, considering the relatively short observation period and the typically prolonged onset of MRONJ, further long-term randomized controlled trials are necessary to accurately assess the effects of dosing frequency on MRONJ incidence [37].

Adjusting denosumab dosing intervals has been proposed as a strategy for reducing the incidence of MRONJ; however, current evidence remains inconsistent, highlighting the necessity for prospective randomized trials. Furthermore, multidisciplinary collaboration between oncologists, dentists, and oral surgeons is essential for optimal patient outcomes. Further studies should refine the MRONJ risk stratification models and evaluate targeted preventive interventions to enhance patient care.

### 4.5. Study Limitations and Future Directions

This study has several limitations that should be considered when interpreting the results. First, the retrospective nature of the study introduces potential selection bias, which limits the ability to establish causality. Although our findings suggest a strong association between cumulative denosumab administration and MRONJ risk, prospective randomized controlled trials are necessary to confirm these results.

Second, the study was conducted at a single institution, which may limit the generalizability of the findings to broader populations. Differences in clinical practice, patient demographics, and dental management protocols across institutions may influence the incidence of MRONJ. Future multicenter studies with larger sample sizes will help validate our findings and improve external validity.

Third, this single-center study did not include a bisphosphonate-treated comparator arm. However, contemporary large-scale studies have reported lower MRONJ rates in zoledronic acid cohorts [20,21], supporting the relative risk elevation observed with denosumab. Validation in multicenter and non-Japanese populations is also warranted to confirm external generalizability.

Fourth, although we accounted for several known MRONJ risk factors, we were unable to comprehensively assess dental-related variables, such as periodontal disease status, prior dental extractions, or denture use. These factors have been suggested to play a crucial role in the development of MRONJ, and their impact should be further investigated in future studies.

Finally, although a significant threshold for cumulative denosumab exposure (32 doses) was identified, the optimal risk stratification strategy remains unclear. Future research should integrate additional risk factors into predictive models to improve MRONJ prevention.

## 5. Conclusions

In this retrospective cohort of 324 patients with breast cancer and bone metastases treated with denosumab, 31.2% had MRONJ. The multivariate logistic regression analysis revealed that cumulative denosumab dose (OR: 1.047 per administration; 95% CI: 1.033–1.061; *p* < 0.001) and history of tooth extraction (OR: 4.40; 95% CI: 2.23–8.71; *p* < 0.001) were the only independent predictors of MRONJ development, whereas diabetes mellitus, hormone therapy, and other systemic factors did not remain significant. ROC analysis yielded an AUC of 0.83 (95% CI: 0.78–0.88) with an optimal cutoff value of 32 cumulative doses, providing a clinically actionable threshold. These findings highlight the necessity of pretreatment dental evaluation and individualized risk stratification—particularly for patients undergoing prior tooth extractions and those approaching 2 years of denosumab therapy. Prospective studies are necessary to validate these thresholds and assess the impact of preventive strategies, such as intensified oral hygiene protocols and optimized dosing intervals, on reducing the incidence of MRONJ.

## Figures and Tables

**Figure 1 cancers-17-02242-f001:**
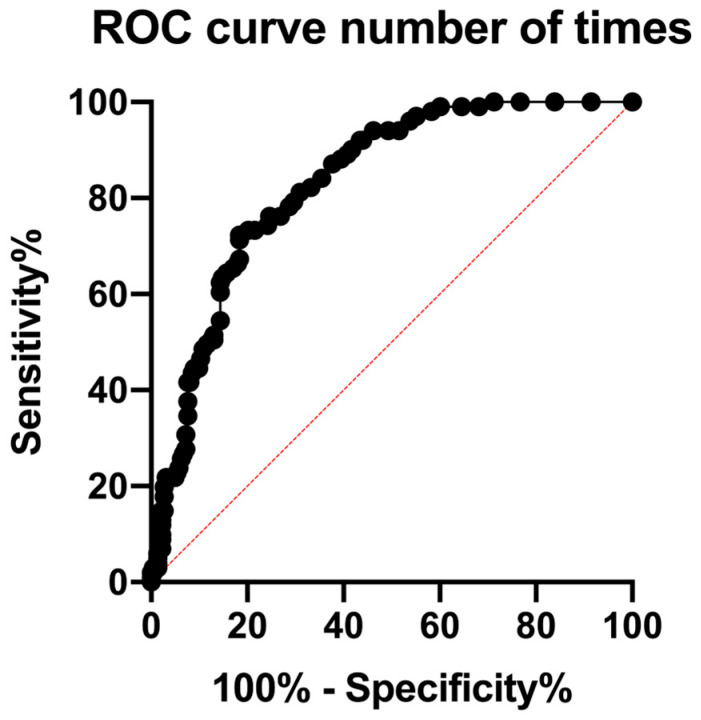
Receiver operating characteristic (ROC) curve for cumulative denosumab doses predicting MRONJ occurrence. The ROC curve assessing cumulative denosumab doses as a predictor of MRONJ incidence yielded an area under the curve (AUC) of 0.83 (95% CI: 0.78–0.88, *p* < 0.0001). The red dashed diagonal line represents the reference line (AUC = 0.5), indicating no discriminative ability. The optimal cutoff for cumulative doses was identified as 32. Patients receiving more than 32 doses exhibited a significantly increased risk of developing MRONJ.

**Figure 2 cancers-17-02242-f002:**
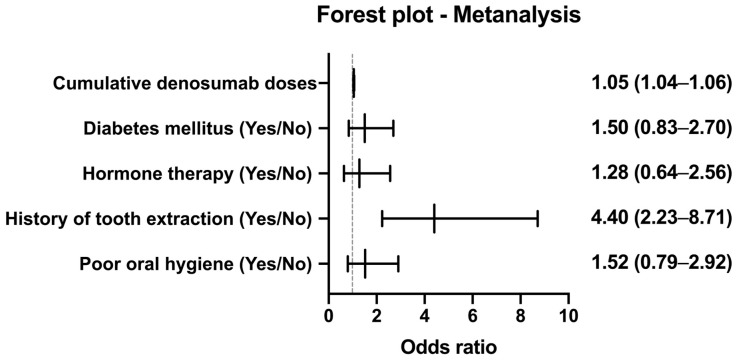
Cumulative MRONJ-free survival by denosumab exposure. Forest plot showing odds ratios (ORs) and 95% confidence intervals (CIs) for the risk of medication-related osteonecrosis of the jaw (MRONJ), based on the multivariable logistic regression analysis (see Table 3). Variables include cumulative denosumab doses (treated as a continuous variable), history of tooth extraction, poor oral hygiene, diabetes mellitus, and hormone therapy. The vertical reference line at OR = 1 indicates no effect.

**Figure 3 cancers-17-02242-f003:**
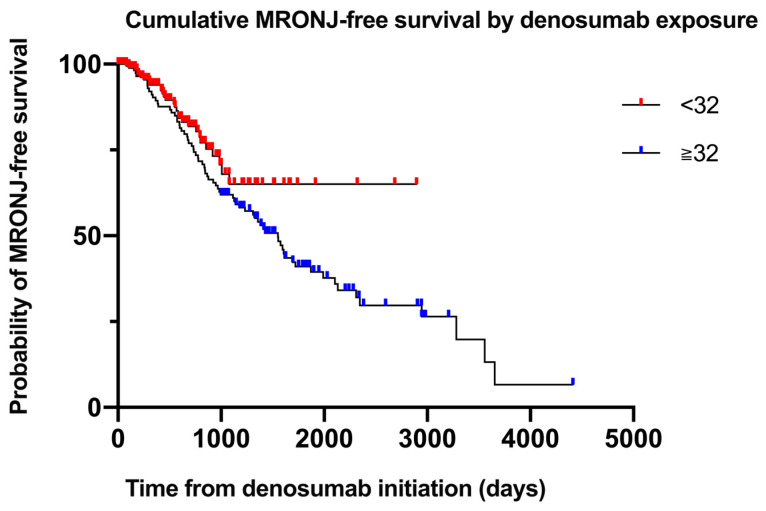
Cumulative MRONJ-free survival by denosumab exposure. Kaplan–Meier curves illustrating MRONJ-free survival stratified by cumulative denosumab dose (<32 vs. ≥32 doses). Numbers at risk are shown at 0, 12, 24, 36, and 48 months.

**Table 1 cancers-17-02242-t001:** Patient demographics. Baseline characteristics of 324 patients with breast cancer and bone metastases included in this study. Data are presented as median (range) for continuous variables and as percentages for categorical variables.

Variable	Median (Range) or %
Age (years)	61.6 (29–100)
Cumulative denosumab doses	21 (1–133)
Incidence of MRONJ	31.2% (n = 101/324)
Diabetes mellitus	47.2%
Smoking history	18.2%
Cumulative prednisolone dose (mg)	1342 (0–24,744)
Chemotherapy	70.4%
Hormone therapy	74.1%
Molecular targeted therapy	80.2%

Abbreviation: MRONJ, medication-related osteonecrosis of the jaw.

**Table 2 cancers-17-02242-t002:** Univariate analysis. Comparison of clinical and treatment-related variables between patients who developed medication-related osteonecrosis of the jaw (MRONJ) (n = 101) and those who did not (n = 223). Data are presented as median (range) for continuous variables and as counts (Yes/No) for categorical variables. *p*-values were calculated using the Mann–Whitney U test for continuous variables and Fisher’s exact test for categorical variables.

Variable	MRONJ (n = 101)	Non-MRONJ (n = 223)	*p*-Value
Cumulative denosumab doses	42 (5–133)	12 (1–110)	<0.001
Age (years, median (range))	64 (39–87)	61 (29–100)	0.991
Diabetes mellitus (Yes/No)	62/39	91/132	<0.001
Smoking history (Yes/No)	14/87	45/178	0.214
Chemotherapy (Yes/No)	69/32	159/64	0.601
Hormone therapy (Yes/No)	83/18	157/66	0.029
Molecular targeted therapy (Yes/No)	81/20	179/44	>0.999
Bevacizumab (Yes/No)	47/54	94/129	0.471
Everolimus (Yes/No)	13/88	40/183	0.331
History of tooth extraction (Yes/No)	44/57	26/197	<0.001
Poor oral hygiene (Yes/No)	44/57	45/178	<0.001
Cumulative prednisolone dose (mg)	1408 (0–24,744)	1319 (0–13,014)	0.808

Abbreviation: MRONJ, medication-related osteonecrosis of the jaw.

**Table 3 cancers-17-02242-t003:** Receiver operating characteristic (ROC) analysis of cumulative denosumab dose as a predictor of MRONJ incidence (n = 324). Data are presented as the area under the curve, standard error, 95% confidence interval, and *p*-value.

Metric	Area
Area under the curve	0.83
Standard error	0.02
95% confidence interval	0.78–0.88
*p*-value	<0.0001

Abbreviation: MRONJ, medication-related osteonecrosis of the jaw.

**Table 4 cancers-17-02242-t004:** Multivariate logistic regression analysis. Results of multivariate logistic regression identifying independent risk factors for medication-related osteonecrosis of the jaw (MRONJ). Odds ratios (ORs) and 95% confidence intervals (CIs) are provided for each variable. Cumulative denosumab doses and history of tooth extraction were identified as statistically significant risk factors (*p* < 0.05), while diabetes mellitus, hormone therapy, and poor oral hygiene were not significant after adjustment.

Variable	OR	95% CI (Lower–Upper)	*p*-Value
Cumulative denosumab doses	1.047	1.033–1.061	<0.001
Diabetes mellitus (Yes/No)	1.500	0.834–2.697	0.176
Hormone therapy (Yes/No)	1.279	0.639–2.562	0.487
History of tooth extraction (Yes/No)	4.402	2.225–8.711	<0.001
Poor oral hygiene (Yes/No)	1.517	0.792–2.907	0.209

**Table 5 cancers-17-02242-t005:** Log-rank test comparing MRONJ-free survival between denosumab exposure groups (n = 324). The log-rank *p*-value for the difference in survival curves (<32 vs. ≥32 doses) is presented.

Log-Rank Test	
*p*-value	0.071

**Table 6 cancers-17-02242-t006:** Numbers at risk at 0, 12, 24, 36, and 48 months by cumulative denosumab dose group. Data represent counts of patients remaining at risk in <32-dose and ≥32-dose cohorts at each time point.

Months	<32 Doses	≥32 Doses
0	211	113
12	114	102
24	59	86
36	23	66
48	11	45

## Data Availability

The datasets generated and analyzed during the current study are available from the corresponding author upon reasonable request.

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
