# Peer review of "Denosumab Dosage and Tooth Extraction Predict Medication-Related Osteonecrosis of the Jaw in Patients with Breast Cancer and Bone Metastases"

_cancers, 2025, doi:10.3390/cancers17132242_

Round 1
Reviewer 1 Report
Comments and Suggestions for Authors
The manuscript entitled “Denosumab Dosage and Tooth Extraction Predict Medication-Related Osteonecrosis of the Jaw in Patients with Breast Cancer and Bone Metastases” by Ohara Sakiko’s group addresses an important clinical issue regarding identification of risk factors and a dose threshold for medication-related osteonecrosis of the jaw (MRONJ) in breast cancer patients treated with denosumab.
The amount of work done in this study is impressive. However, some minor corrections should be made before the manuscript is accepted for publication.
This study claims to address a gap regarding the cumulative dose threshold. However, in the introduction, they should clearly emphasize how their work differs from prior studies, including the meta-analyses. What was the rationale behind 32 doses, and whether this idea has been previously suggested or is entirely novel?
For their statistical analysis, there is no mention of internal validation for the predictive model or ROC threshold. In addition, the relatively high incidence of MRONJ (31.2%) is much higher than in many previous studies. The authors should discuss possible reasons for this discrepancy and possibly discuss the statistical power and sample size justification.
Author Response
Please see the attached Word file for our detailed point-by-point responses to the reviewers’ comments.

Reviewer 2 Report
Comments and Suggestions for Authors
Appreciate the chance to read this. The topic makes sense and feels timely, especially with how often denosumab is used. I liked the idea of trying to pin down a threshold dose for MRONJ — that’s a useful angle.
That said, a few parts left me with questions. I wasn’t entirely sure how the variables were picked for the multivariable model, and I didn’t see anything on collinearity checks — maybe I missed it? Also, dental info is mentioned, but nothing specific about periodontal disease, dentures, etc., which I think would matter here. Feels like that’s a gap.
The bit about the Cox model vs. logistic results — yeah, you mention it, but I’d probably say more about that, even just to offer some possible reasons for the mismatch. Early censoring maybe, but could also be something else?
And no bisphosphonate group — would’ve helped to have that for context. Also, since it's all from one center in Japan, maybe say a word about how well this would hold up in other settings?
Anyway, overall, it’s a solid study. Just needs a bit more clarity in places.
Author Response

(The authors gave the same response as above.)

Reviewer 3 Report
Comments and Suggestions for Authors
Even though the study is a retrospective one, the resulkt and the analyses are very interesting. I further encourage the develope of a prospective study.
I also wonder if you thought about the role of any radiation in the area (for both palliative for pain or ablative reason for oligoprogression)?
I think it would be interesting.
Author Response

(The authors gave the same response as above.)
